# Characteristic Evaluation of Gel Formulation Containing Niosomes of Melatonin or Its Derivative and Mucoadhesive Properties Using ATR-FTIR Spectroscopy

**DOI:** 10.3390/polym13071142

**Published:** 2021-04-02

**Authors:** Prangtip Uthaiwat, Aroonsri Priprem, Ploenthip Puthongking, Jureerut Daduang, Chatchanok Nukulkit, Sirinart Chio-Srichan, Patcharee Boonsiri, Suthasinee Thapphasaraphong

**Affiliations:** 1Graduate School, Khon Kaen University, Khon Kaen 40002, Thailand; phangtip_11@hotmail.com; 2Faculty of Pharmacy, Mahasarakham University, Maha Sarakham 44150, Thailand; aroonsri@kku.ac.th; 3Melatonin Research Group, Khon Kaen University, Khon Kaen 40002, Thailand; pploenthip@kku.ac.th; 4Department of Pharmaceutical Chemistry, Faculty of Pharmaceutical Sciences, Khon Kaen University, Khon Kaen 40002, Thailand; 5Centre for Research and Development of Medical Diagnostic Laboratories, Faculty of Associated Medical Sciences, Khon Kaen University, Khon Kaen 40002, Thailand; jurpoo@kku.ac.th; 6Department of Thai Traditional Medicine, Rajamangala University of Technology Isan Sakon Nakhon Campus, Sakon Nakhon 47160, Thailand; chatnuk@gmail.com; 7Synchrotron Light Research Institute (Public Organization), Nakhon Ratchasima 30000, Thailand; sirinart@slri.or.th; 8Department of Biochemistry, Faculty of Medicine, Khon Kaen University, Khon Kaen 40002, Thailand; patcha_b@kku.ac.th

**Keywords:** gel base effect, mucosa, ATR-FTIR spectroscopy, melatonin niosome gel

## Abstract

Chitosan or polyvinyl pyrrolidone (PVP) were used in combination with hydroxypropyl methylcellulose (HPMC) and poloxamer 407 (P407) as gelling agents for oral drug delivery. The performance interaction with mucin of chitosan-composed gel (F1) and PVP-composed gel (F2) was compared using attenuated total reflectance–Fourier-transform infrared (ATR-FTIR) spectroscopy at controlled temperatures of 25 and 37 °C for 1 and 5 min. F1 containing niosome-entrapped melatonin or its derivatives was investigated for mucoadhesive interaction on mucosa by ATR-FTIR spectroscopy under the same conditions. The results showed that F1-treated mucin gave a significantly lower amide I/amide II ratio than untreated mucin and F2-treated mucin did within 1 min, suggesting improved rapid affinity between mucin and chitosan. The spectra of mucosa treated with F1 incorporating niosomes of melatonin or its derivatives showed peak shifts at C=O (amide I), N-H (amide II), and carbohydrate regions and an associated decrease in the amide I/amide II ratio and increase in the carbohydrate/amide II ratio. These results indicated electrostatic interaction and hydrogen bonding between chitosan and mucin on the mucosa. In conclusion, the molecular interaction between gels and mucin/mucosa detected at amide I and amide II of proteins and the carbohydrate region could lead to an improved mucoadhesive property of the gel on the mucosa.

## 1. Introduction

Several mucoadhesive polymers including hydroxypropyl methylcellulose (HPMC), polyvinylpyrrolidone (PVP), carbopol, chitosan, carboxymethylcellulose, and poloxamer 407 (P407) have been utilized in controlled-release drug delivery systems. HPMC, a hydrophilic and non-ionic cellulose [1,2] with high swelling and bioadhesive properties [3], has been used in combination with P407 and PVP to form a mucoadhesive gel incorporating melatonin (MLT)-encapsulated niosomes, providing transmucosal delivery and prolonged release of MLT in the oral cavity [4]. Chitosan, a cationic polyaminosaccharide deacetylated from chitin, has been shown to interact with negatively charged mucin and enhance penetration through epithelial tight junctions [5,6,7]. Furthermore, chitosan can stabilize and synergize the antioxidant activity of melatonin [8].

Exogenous MLT has poor oral bioavailability (9–33%) and a short half-life [9]. Encapsulation of MLT in niosomes has been shown to prolong its time in systemic circulation [4,10,11], and recently, two MLT derivatives with longer releasing properties, namely succinyl-MLT (S-MLT) and glutaryl-MLT (G-MLT), were synthesized [12].

Adhesion to mucosal membranes is one of the key factors for successful transmucosal delivery of drugs [13]. In the oral cavity, there are additional challenges from the movement of the oral mucosa, saliva secretion, and drainage via swallowing. Mucoadhesive gels can provide advantages in transmucosal drug delivery by prolonging contact time, enhancing drug absorption, bypassing first-pass metabolism, and preventing drug loss [14]. Mucin is the principal glycoprotein component of mucus, which coats the mucosa and is an important target for interaction with mucoadhesive gels. The polypeptide chains of mucin have domains rich in threonine and/or serine that compose hydroxyl groups linked with negatively charged oligosaccharides [15]. Currently, there is no single method available for in vitro evaluation of the mucoadhesive properties of a drug delivery system, although one of the most used parameters is the measurement of the mechanical forces required to detach a testing gel from an excised piece of mucosa by a texture analyzer [16,17]. However, mechanical parameter testing has shown some inconsistencies and low discrimination power, which limits its translation to real situations [18].

Attenuated total reflectance–Fourier-transform infrared (ATR-FTIR) spectroscopy, a powerful tool for determination of small changes in biochemical interactions, is used to distinguish heterogeneous functional groups in biological tissue samples [19]. Chain interpenetration of polyacrylic acid at the mucin interface was interpreted by using ATR-FTIR spectroscopy to explain the mucoadhesive mechanism [20]. Studies investigated mucin–mucoadhesive polymer interaction by FTIR spectroscopy and found the main mechanisms to be hydrogen bonding and disulfide bridges [21,22].

Using ATR-FTIR spectroscopy, we compared the in vitro mucoadhesive properties of our previous transmucosal gel formula containing HPMC, P407, and PVP with a gel formula containing HPMC, P407, and chitosan. ATR-FTIR spectroscopy is a convenient technique to evaluate the mucoadhesive property of polymers by analysis of interfacial interaction or interpenetration between a polymer and mucin. This technique is less time-consuming and easy to perform without any complicated procedure. Therefore, ATR-FTIR spectroscopy is a suitable technique for screening the mucoadhesive properties of polymers for mucoadhesive drug delivery systems.

## 2. Materials and Methods

### 2.1. Materials

HPMC (Onimax Co. Ltd., Bangkok, Thailand), PVP (Dai-ichi Kogyo Seiyaku, Kyoto, Japan), MLT (Shanghai Chemical, Shanghai, China), sodium chloride (Carlo Erba, Milano, Italy), sorbitan monostearate (Span60), cholesterol, P407, chitosan from shrimp shells, mucin from porcine stomach, succinic anhydride, glutaric anhydride and sodium hydride (60% in oil) (Sigma-Aldrich, St. Louis, MO, USA), dimethylformamide, methanol, chloroform, n-hexane, hydrochloric acid, ethyl acetate and dichloromethane (Labscan, Bankok, Thailand), and normal saline solution (NSS, A.N.B. Laboratories, Bankok, Thailand) were used as received.

### 2.2. Synthesis of MLT Derivatives

The S-MLT and G-MLT derivatives were synthesized as described previously, with modifications [12]. Briefly, sodium hydride (3 mmol) was dissolved with 3 mL of n-hexane in a 50-mL round-bottom flask, vacuumed, and reacted with MLT (2 mmol), which was dissolved in 5 mL of dimethylformamide. The mixture was stirred at room temperature (RT) for 30 min and placed in an ice bath (0–5 °C). Succinic anhydride or glutaric anhydride (2.2 mmol) was added into the mixture and stirring was continued at RT for 24 h. The mixture was then cooled in an ice bath for 10 min, followed by adjusting the pH to 2 by using a 5% HCl solution. The precipitate was collected and purified using column chromatography. The gradient system started from 10% ethyl acetate and 90% hexane and the ratio of ethyl acetate was increased until reaching 100% ethyl acetate. After that, a gradient of 95% ethyl acetate and 5% methanol was used and the ratio was increased to 80% ethyl acetate and 20% methanol. The thin-layer chromatography (TLC) method was applied to monitor the purity of the MLT derivatives. The products were spotted in a TLC chamber with a solvent system (10% methanol: 90% dichloromethane). The TLC plate was air-dried and observed under UV light at 254 and 365 nm. 

### 2.3. Mucoadhesive Gel Formulation

The gel bases were modified from a previous study [4]. Gel formulation 1 (F1) was composed of HPMC (8.7% *w/w*), P407 (0.4% *w/w*), and chitosan (0.12% *w/w*). Gel formulation 2 (F2) was composed of HPMC (8.7% *w/w*), P407 (0.4% *w/w*), and PVP (0.12% *w/w*). HPMC, P407, and PVP were prepared by dissolving in deionized water. Chitosan was dissolved in sodium acetate-acetic acid buffer at pH 4.2 (0.2 M acetic acid and 0.2 M sodium acetate). Niosomes (2% *w/w*) containing MLT or its derivatives were prepared using a thin film method in which 0.2 g each of Span60, cholesterol, and MLT or its derivatives was added in a 10-mL round-bottom flask with 5 mL of chloroform-methanol. Then, the mixture was sonicated in a water bath sonicator (Kudos SK3210HP, Shanghai, China) at 60 °C for 10 min. The solvent was removed at 55 °C under 200 mbar by a rotary evaporator (Buchi, Switzerland) for 5 min to produce a thin film. The F1 gel acted as a vehicle in the gel base preparation phase. Thus, the liquid dispersion of niosomes was then mixed with the F1 gel until the final weight was 10 g. A blank niosome in F1 gel was prepared using the same protocol but without MLT or its derivatives. 

### 2.4. Niosome Gel Characterization

The zeta potential and the polydispersity index (p.i.) of niosome gels were assessed using a nanoparticle size analyzer (SZ-100 HORIBA, Kisshoin Minami-Ku, Kyoto, Japan). Each sample was diluted to 1:10,000 in deionized sterile water and sonicated for 30 min before measurement. All measurements were performed in triplicate. The morphology of niosome gels was evaluated by transmission electron microscopy (TEM, FEI Company, Hillsboro, ORE, USA) and an analyzer (Perkin Elmer model 1022 LC plus, West Berlin, NJ, USA). Each sample was sonicated for 30 min before dropping onto a copper grid. Then, the sample was dried at RT for 24 h before measurement. The vesicle sizes of niosomes were measured and averaged from ten niosomes in each sample using the Image J program.

### 2.5. Encapsulation Efficiency of MLT and Its Derivatives in Niosome Gel

The encapsulation efficiency of MLT, S-MLT, or G-MLT in the niosome gel was evaluated after ultracentrifugation at 15,000 rpm at 4 °C for 30 min; then, the liquid supernatant was separated from the sediment. The absorbances of the supernatant and the sediment were determined at 277 nm for MLT and 255 nm for its derivatives by UV spectrometry. The encapsulation efficiency was calculated as follows:(1)$$\%\;\mathrm{Encapsulation}\;\mathrm{efficiency}\frac{\mathrm{Amount}\;\mathrm{of}\;\mathrm{drug}\;\mathrm{in}\;\mathrm{sediment}}{\mathrm{Amount}\;\mathrm{in}\;\mathrm{supernatant}+\mathrm{amount}\;\mathrm{in}\;\mathrm{sediment}}\;\times \;100$$

### 2.6. Texture Analysis

The texture profile analysis was modified from a previous study [23] to assess the detachment forces and work of adhesion of gels (blank niosomes in F1, MLT niosomes in F1, S-MLT niosomes in F1, and G-MLT niosomes in F1) with mucosa. This experiment was performed using a texture analyzer (TA-XT Plus, Stable Micro Systems, Godalming, UK) and a mucoadhesive rig (10 mm in diameter, AMUC, Stable Micro Systems, Godalming. UK). The rig was fixed with a piece of porcine esophagus mucosa (2 × 2 cm), which was collected from a local slaughterhouse after sacrifice. The porcine esophagus was immersed in NSS at 60 °C for 1 min to separate its connective tissue. The excised mucosa was stored at −20 °C for use within 1 week. Then, 0.1 g of each gel was placed onto the probe, 10 mm in diameter. The mucosa, with a contact area of 0.785 cm^2^, was compressed/decompressed at a rate of 6 mm s^−1^ of the test speed, followed by compression for a 15-s delay. The work of adhesion was obtained from the area under the curve between the force and distance during decompression. Each sample was tested with six experiments at room temperature.

### 2.7. Permeation Test

The porcine esophagus mucosa was placed between donor and receptor compartments of a 5-mL Franz diffusion cell (Crown Glass Company, Jersey City, NJ, USA.). The mucosa was immersed in deionized (DI) water, pH 7.4, for 15 min before use. A 0.5-g sample was spread by using a syringe and a glass stirring rod onto the mucosa at the donor side. The receptor compartment was filled with DI water, pH 7.4, which was stirred at 60 rpm, and the cell was incubated at 37 ± 1 °C. A 0.5-mL sample was removed from the receptor compartment and replaced with the same volume of fresh solution at 0, 1, 3, 6, 9, 12, 15, 18, and 24 h. The amount of MLT and its derivatives in the receptor fluid was determined by HPLC analysis. For HPLC conditions, a Luna 5U PEP (2) 100A, 250 × 4.6-cm (Phenomenex, Torrance, CA, USA) column was used in this experiment. The mobile phase was composed of phosphate buffer, pH 7.2 (0.4% *w/v* NaH_2_PO_4_.2H_2_O and 0.1% *w/v* Na_2_HPO_4_), and acetonitrile (75:25 *v/v*), with a flow rate 1.0 mL/min. MLT, S-MLT, and G-MLT were detected using a UV detector at 220 nm.

### 2.8. Mucin Treatment

F1- or F2-treated mucin was analyzed by ATR-FTIR spectroscopy with parameters modified from the previous method [21]. A 5% *w/w* mucin solution in deionized water was incubated at 25 ± 2 (RT) or 37 ± 2 °C (body temperature, BT). The mucin solution was transferred to the crystal sample holder of the ATR-FTIR spectroscope. An equal volume of F1 or F2 was dropped into the mucin solution. After incubation for 1 and 5 min, the ATR-FTIR spectra were recorded. The interaction between mucin and F1 or F2 was evaluated from the ATR-FTIR spectra. Untreated mucin was the negative control. 

### 2.9. Mucosa Treatment

The mucosal sheet was immersed in NSS at 37 °C for 15 min before use [24]. The mucosal sheet was cut into 0.5 × 0.5-cm^2^ pieces and mounted with 0.01 mL of NSS to ensure full hydration before the gel application. The mucosal surface of each sample was pretreated with 20 mg each of the gels (F1, blank niosomes in F1, MLT niosomes in F1, S-MLT niosomes in F1, and G-MLT niosomes in F1) at RT and BT for 1 and 5 min and then scraped 3 times with a plastic scraper. 

The surface of the treated mucosa was pressed onto ATR diamond crystal for FTIR spectral acquisition. Samples were examined in triplicate for FTIR spectra. The untreated mucosa was used as a negative control. The spectra of blank niosomes in F1, MLT niosomes in F1, S-MLT niosomes in F1, G-MLT niosomes in F1, MLT, S-MLT, and G-MLT were recorded. 

### 2.10. ATR-FTIR Spectroscopic Analysis

ATR-FTIR spectra were recorded by an ATR-FTIR spectrometer (Agilent technologies 4500 series FTIR, Kuala Lumpur, Malaysia). The spectra were obtained in the range of 4000–650 cm^−1^ and at 4 cm^−1^ resolution with 256 co-added scans per spectrum. Each spectrum was normalized and integrated by OPUS 7.2 (Bruker, Hanau, Germany). The absorption intensities under the spectra were integrated at wavenumbers of 1695–1596, 1596–1493, and 1189–973 cm^−1^, which represent the regions of amide I, amide II, and carbohydrate, respectively.

Absorption unit ratios of the integrated results were evaluated for amide I/amide II and carbohydrate/amide II ratios from F1- or F2-treated mucosa with or without niosomes in comparison to the relevant blanks.

From interaction and non-interacted mucosa, samples were analyzed by principal component analysis (PCA). Spectral pre-processing was conducted by taking the second derivative, smoothing with a Savitzky–Golay function (3 polynomials and 15 smoothing points), and correcting the spectral scattering at fingerprint (1780–980 cm^−1^) by using Extended Multiplicative Scatter Correction of a computer software (Unscrambler X 10.5, (CAMO Software AS, Oslo, Norway).

### 2.11. Statistical Analysis

The data are reported as mean ± standard deviation (SD) and were evaluated for normal distribution using the Shapiro–Wilk test. The parametric data were analyzed by one-way analysis of variance (one-way ANOVA) and pairwise comparisons were carried out by using Tukey’s penalizations. The non-parametric data analyses were performed with a Kruskal–Wallis one-way analysis of variance and pairwise comparisons were performed using the Mann–Whitney U test. All statistical analyses were performed using SPSS version 19 (IBM Corp, Chicago, IL, USA). The level of significance was *p* < 0.05.

## 3. Results and Discussion

### 3.1. ATR-FTIR Spectra of the Mucin, Mucosa, and Gels

Figure 1a shows the ATR-FTIR spectra of mucin. We observed peaks at 1634 (C=O stretching of amide I), 1553 (C–N stretching of amide II), 1452 (C–H bending), 1375 (CH_3_ bending), 1317 and 1236 (amide III), 1146 and 1117 (C–O stretching), 1077 cm^−1^ (C–N stretching), 1053 cm^−1^ (C–O stretching or C–O bending of carbohydrates), and 970 cm^–1^ (phosphorylated proteins) [25,26,27]. The ATR-FTIR spectrum of mucin showed the peptide bond (amide I and amide II) and oligosaccharide (C–O stretching) parts of the glycoprotein. The ATR-FTIR spectra of porcine esophagus mucosa established molecular vibration at 1644 (C=O stretching of amide I), 1545 (N–H bending and C–N stretching of amide II), 1452 and 1397 (C–H bending), 1236 (amide III), 1146 and 1117 (C–O stretching), 1077 (C–N stretching), and 1053 cm^−1^ (C–O stretching or C–O bending of carbohydrates) [26,27].

The amide I, amide II, and C–H bending of ATR-FTIR spectra of porcine esophagus showed similar molecular vibrational characteristics as those of normal human esophagus [28]. In this study, the ATR-FTIR spectrum of mucosa showed lower intensity of oligosaccharide peaks than mucin (Figure 1a), because oral and esophageal mucosa contain only 0.1–0.5% mucin [29]. Figure 1b displays the ATR-FTIR spectral characteristics of the gels. F1 and F2 exhibited similar ATR-FTIR spectra at 1452, 1375, 1146 and 1117, 1077, 1053, and 946 cm^−1^, representing C–H bending, CH_3_ bending, C–O stretching, C–N stretching, C–O stretching, and O–H bending, respectively [25,26,27]. According to the F1 and F2 spectra, the molecular vibration of gels is similar to these constituent polymers.

The ATR-FTIR spectra of mucosa treated with MLT, S-MLT, or G-MLT in F1 all presented a similar peak at 1018 cm^−1^ (C–O, C–C, and O–CH of polysaccharides). The ATR-FTIR spectra of mucosa treated with blank niosomes, MLT niosomes, S-MLT niosomes, or G-MLT niosomes in F1 exhibited peaks similar to the ATR-FTIR spectrum of F1. However, G-MLT niosomes in F1 showed a peak at 1018 cm^−1^, indicating that side chains of G-MLT might appear outside of the niosome lipid bilayer. There were no peaks at 1018 cm^−1^ in the ATR-FTIR spectra of MLT niosomes in F1 or S-MLT niosomes in F1, which suggests that all parts of MLT and S-MLT were encapsulated into the bilayer of the niosome (% entrapment efficiency > 90%).

### 3.2. F1- and F2-Treated Mucin

From the ATR-FTIR spectra of samples incubated at RT for 1 or 5 min, the average absorption ratio of amide I to amide II in untreated mucin was around 1.0. F1-treated mucin provided an average amide I/amide II ratio lower than that of untreated mucin and F2 gel-treated mucin at both 1 and 5 min (*p* < 0.05, Figure 2). These changes in the amide I/amide II ratio indicate that F1 interacted with mucin and F2 did not. This difference could be an effect of the chitosan in F1. The cationic chitosan could provide an electrostatic interaction with the negative charge of sialic acid in mucin, promoting strong mucoadhesion [30,31]. In addition, the strong interaction between mucin and chitosan in F1 causes the protection of the peptide bond in mucin from the interface [31]. Therefore, a reduction in the amide I/amide II ratio was observed from the ATR-FTIR spectrum of F1-treated mucin. The observed reduction in the amide I/amide II ratio in F1-treated mucin could also be caused by hydrogen bonding between the NH_2_ of chitosan and the COOH of sialic acid in mucin. In contrast, F2 is a non-ionic polymer that would have no electrostatic interaction with mucin. This strong mucoadhesive property of F1 makes it a promising gel formulation for incorporation of MLT, S-MLT, and G-MLT niosomes.

Figure 3 shows the absorption unit ratios of F1-treated mucin to untreated mucin for amide I/amide II and carbohydrate/amide II. An absorption unit ratio of 1 represents no difference between treated mucin and untreated mucin.

At RT, the F1-treated mucin/untreated mucin absorption ratio for amide I/amide II was less than one, with no significant difference between 1 and 5 min (*p* > 0.05), whereas the absorption unit ratio for carbohydrate/amide II was higher than one, with no significant difference between 1 and 5 min (*p* > 0.05).

At BT, the amide I/amide II absorption ratio of F1-treated mucin/untreated mucin was less than one. This indicates that the amide I/amide II ratio of mucin was reduced after treatment with F1. At BT, there were no significant differences (*p* > 0.05) between interactions at 1 and 5 min for both amide I/amide II and carbohydrate/amide II. However, for carbohydrate/amide II, the F1-treated mucin/untreated mucin absorption ratio at BT was higher than that at RT, with a significant difference between 1 and 5 min (Figure 3, *p* < 0.05).

The results show that F1 interaction with mucin could be detected at amide I, amide II, and the carbohydrate region within 1 min and that the interaction was more extensive at BT than it was at RT. This is presumably because the interaction temperature affects the thermosensitive polymer P407, which is in a solution state at RT and a gel state at BT [32].

In this study, we found a reduction in amide I/amide II and an increase in carbohydrate/amide II after mucin interaction with F1. This result was similar to that of Saiano et al. (2003), which showed that the intensities of amide I and amide II were reduced after both α,β-poly(N-hydroxyethyl)-DL-aspartamide polymer and α,β-poly(aspartylhydrazide) polymer interacted with mucin [33].

Moreover, Hsein et al. (2015) reported that amide I and amide II were the important regions for interactions between mucin and reticulated whey protein, forming microparticles. They concluded that hydrogen bonding was established between these two functional groups and mucin at a pH lower than the pKa (2.6) become non-ionized form. At higher pH, mucin became ionized, which caused electrostatic interaction [21].

The absorption unit ratio of carbohydrate/amide II from F1-treated mucin/untreated mucin at both 1 and 5 min was greater than five. This result indicated that F1 interpenetration into mucin could be confirmed by ATR-FTIR spectroscopy in the same way that poly (acrylic acid) interpenetrated into the mucin interface (Jabbari et al. (1993)). They suggested that chain interpenetration across the biointerface was a limitation of adhesion [20]. Similar to Padhye et al. (2017), they reported that the self-adhesion of polymers, HPMC and Polyethylene glycol (PEG), occurred by polymer interpenetration across the interface, leading to bonding [34].

### 3.3. Physical Parameters of Niosome Gels

The physical parameters of blank niosomes, blank niosomes in F1, MLT niosomes in F1, S-MLT niosomes in F1, and G-MLT niosomes in F1 are shown in Table 1. The polydispersity index (p.i.) indicates the homogeneity of the particle suspension. Values should be lower than 0.30, which corresponds to a single size of particle [35]. In this experiment, all niosome gels had p.i. values of more than 0.30, indicating a broad distribution of particle sizes. This is likely to be the effect of the niosome preparation; the thin layer technique we employed has previously been shown to generate a broad range of niosome sizes [36]. The lower negative zeta potential value of blank niosomes in F1 compared to blank niosomes alone (*p* < 0.05) indicates that the chitosan in F1 may have imparted positive charges to the particles. The higher negative zeta potential values of MLT and S-MLT niosomes in F1 compared to blank and G-MLT niosomes in F1 (*p* < 0.05) might be due to a partial insertion of the MLT or S-MLT into the niosome bilayer, with a consequent increase in the negative value of the zeta potential [37]. However, high negative or positive zeta potential values represent the prevention of particle aggregation and indicate increased stability. Thus, MLT and S-MLT niosomes in F1 were more stable than blank and G-MLT niosomes in F1.

The percentage of encapsulation of MLT and its derivatives into niosomes was around 88–93%, which supports the disappearance of the peak at 1018 cm^−1^ in the FTIR spectra of MLT and S-MLT niosomes in F1.

The permeation test was conducted to investigate the permeability of MLT, S-MLT, and G-MLT niosomes in F1 through mucosa after gel adhesion (Table 1). The results show that MLT and its derivatives could penetrate mucosa, but MLT niosomes in F1 had a higher flux rate than S-MLT and G-MLT niosomes in F1 (*p* < 0.05). The S-MLT niosomes in F1 spent more time passing through the mucosa than the other gels (lag time 5 h, *p* < 0.05). Thus, the flux rates and lag times of these gels were different because of the individual properties of MLT and its derivatives.

The relation of detachment force and work of adhesion of blank niosomes in F1, MLT niosomes in F1, S-MLT niosomes in F1, and G-MLT niosomes in F1 represents the mucoadhesive property of the gels, which is shown in Appendix A. The blank and G-MLT niosomes in F1 showed a high relation of detachment force and work of adhesion (*p* < 0.05), with detachment force increasing with the work of adhesion. In contrast, the MLT and S-MLT niosomes in F1 did not have a relation of detachment force and work of adhesion.

The morphology of blank niosomes, blank niosomes in F1 MLT niosomes in F1, S-MLT niosomes in F1, and G-MLT niosomes in F1 is shown in Figure 4. All niosome gels were spherical in shape. The blank and G-MLT niosomes in F1 had a gel nest formation between the particles. The MLT and S-MLT niosomes in F1 had gel formation surrounding the niosome particles. These morphologies correlate with the zeta potential results; the high negative values of MLT 1 and S-MLT niosomes in F1 showed no aggregation. Nevertheless, all samples except blank niosomes showed gel surrounding the niosomes, indicating that the niosomes were covered by the F1 gel.

### 3.4. F1, Blank Niosomes in F1, and MLT or Its Derivative Niosomes in F1-Treated Mucosa

The principal component (PC) plot (Figure 5a) shows the separation of F1-treated mucosa, blank niosomes in F1-treated mucosa, MLT niosomes in F1-treated mucosa, S-MLT niosomes in F1-treated mucosa, and G-MLT niosomes in F1-treated mucosa from untreated mucosa at the fingerprint (1780–980 cm^−1^) along PC-1 with 68% explained variance. The negative loading plot (Figure 5b) shows the spectral absorption of untreated mucosa. The positive loading plot (Figure 5b) shows absorption peak shifts of F1-treated mucosa, blank niosomes in F1-treated mucosa, MLT niosomes in F1-treated mucosa, S-MLT niosomes in F1-treated mucosa, and G-MLT niosomes in F1-treated mucosa from untreated mucosa.

The peak shifts were 1660 (C=O stretching of amide I), 1527 (N–H bending of amide II), 1373 (C–H bending), 1150 (C–O stretching), 1117 (C–OH stretching), 1075 (PO_2_^−^ symmetric stretching), and 1051 cm^–1^ (C–O stretching), involving protein, lipids, phosphate, and oligosaccharides or carbohydrates. This result shows that blank and MLT or its derivative niosomes in F1 could interact with mucosa similarly to F1-treated mucosa.

The major interactions between these gels and mucosa occurred at amide I, amide II, and the carbohydrate region. A previous study reported that reticulated whey protein microparticles (whey protein is a mucoadhesive polymer) interacted with mucin and showed peak shifts at carbonyl C=O and amine N–H. They suggested that the interaction of that region with mucin was established by hydrogen bonding [21]. An illustration of electrostatic interaction between chitosan and mucin is shown in Figure 6a. In our study, the most important shifts were also at amide I and amide II, indicating that the polymers (especially chitosan) surrounding the niosomes could interact with mucin on the mucosa by hydrogen bonding, as shown in Figure 6b. At neutral gel pH, hydrogen bonding between the hydrogen atoms from NH_2_ in chitosan and the oxygen atoms from COOH or OH in the core protein of mucin was expected to be the main interaction. Therefore, the amide I of protein might be protected from the interface, which causes the reduction of amide I or amide I/amide II in the ATR-FTIR spectrum. This is similar to a previous study that concluded that chitosan-coated niosomes interacted with mucin via chitosan polymer interaction with mucin in the mucus layer [38]. The electrostatic interaction might occur less than hydrogen bonding does because the amino groups in chitosan exist in the ionic form (NH_3_^+^) in acidic environments [30]. In contrast, our study used a neutral environment that showed less NH_3_^+^ formation, indicating that electrostatic interactions might occur less than hydrogen bonding does in the chitosan and mucin interaction. Moreover, the other peak shift in the carbohydrate area shows the spectra absorption of the gel base. The carbohydrate peak shifts might represent the chain interpenetration of the major component of the gel base, HPMC, onto mucosa. This was previously suggested by Bravo-Osuna et al. (2012), who postulated that the bioadherence of HPMC occurred from chain interdiffusion into mucin [3]. Esophageal mucosa was shown to contain levels of mucin measured from saliva secretion (0.88 ± 0.11 mg/mL) [29]; strong mucin–polymer affinity provides mechanical protection that is beneficial for mucosal damage.

S-MLT niosomes in F1-treated mucosa showed differences from the other gel-treated mucosa. This may be because S-MLT has a slightly different structure to G-MLT with less CH_2_ (Figure 7a). After separated analysis, S-MLT niosomes in F1-treated mucosa showed discrimination from G-MLT niosomes in F1-treated mucosa along PC-1 with 60% explained variance (Figure 5c). S-MLT niosomes in F1-treated mucosa had a major peak shift from G-MLT-treated mucosa at 1045 cm^−1^ (C–O stretching) in a negative loading plot (Figure 5d), whereas G-MLT niosomes in F1-treated mucosa showed a difference from S-MLT niosomes in F1-treated mucosa at 1018 cm^−1^ (C–O stretching) in a positive loading plot (Figure 5d). We hypothesize that S-MLT niosomes in F1-treated mucosa might have a different interaction at C–O compared to the other compounds. The peak at 1018 cm^−1^ of G-MLT niosomes in F1-treated mucosa might occur from the COOH group of G-MLT, which is extended out of the lipid bilayer (Figure 7c).

### 3.5. Effects of Temperature and Time on Blank Niosomes in F1 and MLT or Its Derivative Niosomes in F1-Treated Mucosa

The amide I/amide II ratios of niosome gel (blank niosomes in F1, MLT niosomes in F1, S-MLT niosomes in F1, and G-MLT niosomes in F1) treated mucosa/untreated mucosa at BT and RT for 1 and 5 min were all lower than one and had no significant differences (*p* > 0.05) between groups (Figure 8). This reduction in the amide I/amide II ratio indicated that all niosome gels could interact at amide I and amide II of mucosa. The interaction occurred rapidly, within 1 min for both temperatures. The niosome gel (blank niosomes in F1, MLT niosomes in F1, S-MLT niosomes in F1, and G-MLT niosomes in F1) treated mucosa/untreated mucosa showed increased carbohydrate/amide II ratios of more than one at both temperatures and at all time points (Figure 9). At RT, all niosome gel-treated mucosae provided no significant differences (*p* > 0.05) between groups at both 1 and 5 min.

At BT, the carbohydrate/amide II ratio presented a ratio higher than one at both 1 and 5 min, with no significant differences (*p* > 0.05) between groups. This result indicated that F1 containing niosomes’ interaction with mucosa could be detected with an increase in the carbohydrate/amide II ratio.

## 4. Conclusions

Chitosan was found to interact with mucin more than PVP did, so chitosan might enhance the mucoadhesive property of gel. The F1 gel (composed of HPMC, chitosan, and P407) showed rapid interaction with mucin, within 1 min. The temperature affected the mucoadhesive interaction of mucin and the F1 gel due to the presence of P407, a thermosensitive gel. The ATR-FTIR results revealed peak shifts of molecular interactions of F1 gel as well as MLT or its derivative niosomes in F1 with the mucosa at C=O (amide I), N–H (amide II), and the carbohydrate region. Moreover, the absorption unit ratios of amide I/amide II for MLT or its derivative niosomes in F1-treated mucosa were reduced and carbohydrate/amide II ratios were increased. Therefore, mucoadhesive interactions between the gels and mucin/mucosa can be investigated from the peak shifts and the changes in absorption unit ratios of ATR-FTIR spectra.

## Figures and Tables

**Figure 1 polymers-13-01142-f001:**
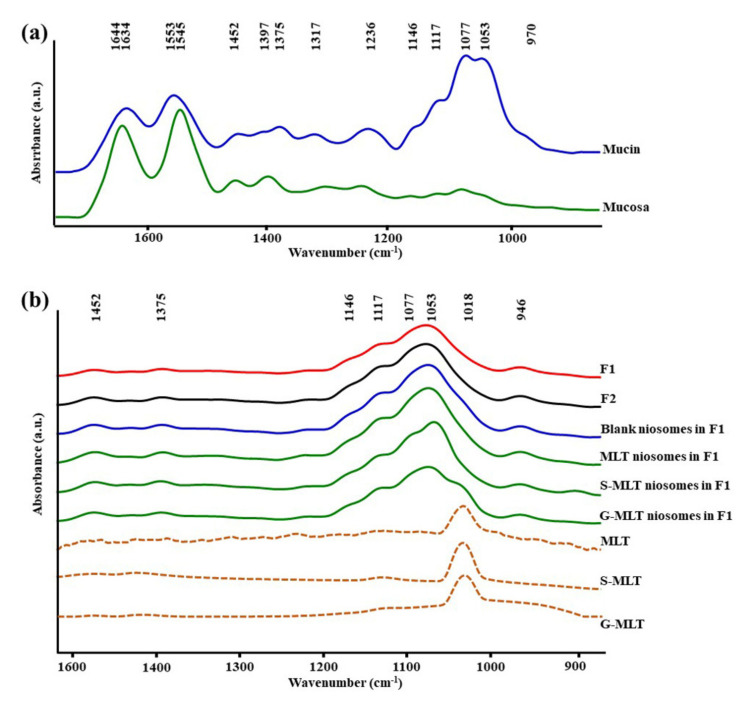
The average attenuated total reflectance-Fourier-transform infrared (ATR-FTIR) spectra of (**a**) mucin and mucosa and (**b**) gel formulation 1 (F1), F2, blank niosomes in F1, melatonin (MLT) niosomes in F1, succinyl-MLT (S-MLT) niosomes in F1, glutaryl-MLT (G-MLT) niosomes in F1, MLT, S-MLT, and G-MLT (*n* = 3).

**Figure 2 polymers-13-01142-f002:**
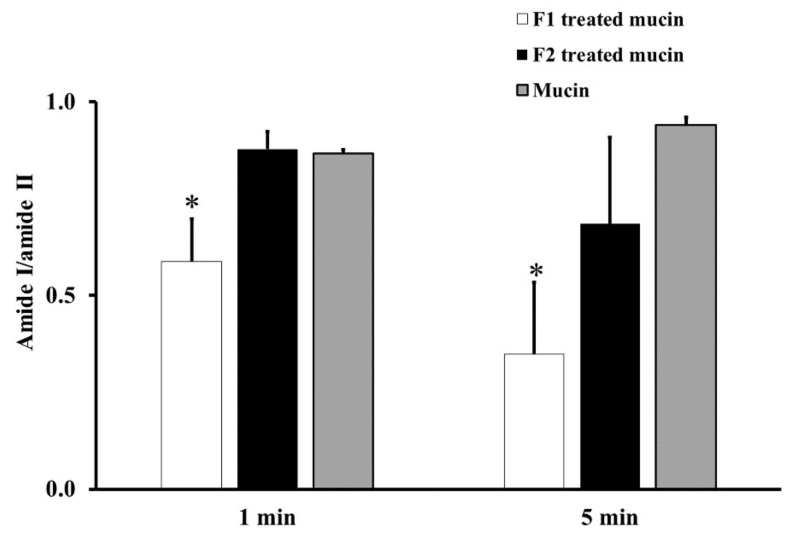
The amide I/amide II absorption ratios at room temperature (RT) for 1 and 5 min of F1-treated mucin (white), F2-treated mucin (black), and untreated mucin (gray). Experiments were performed in triplicate and repeated three times. Bars display mean ± S.D., * *p* < 0.05 compared with untreated mucin.

**Figure 3 polymers-13-01142-f003:**
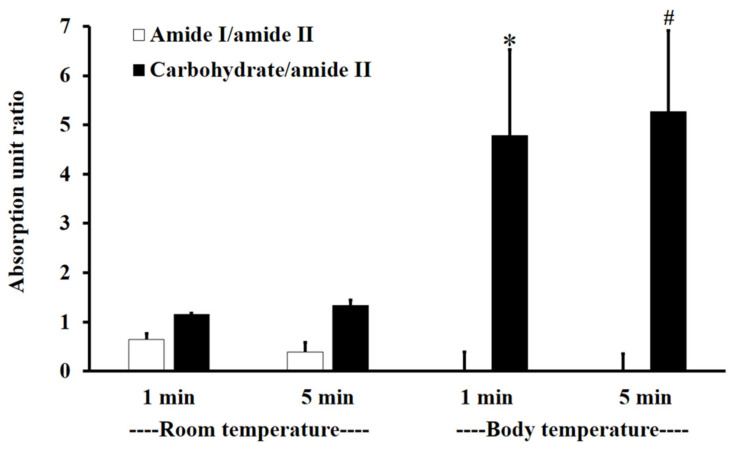
The amide I/amide II absorption unit ratio (white) and carbohydrate/amide II ratio (black) of F1-treated mucin/untreated mucin at body temperature (BT) and RT for 1 and 5 min. Experiments were performed in triplicate and repeated three times. Bars display mean ± S.D., * *p* < 0.05 compared to 1 min at RT, # *p* < 0.05 compared to 5 min at RT.

**Figure 4 polymers-13-01142-f004:**
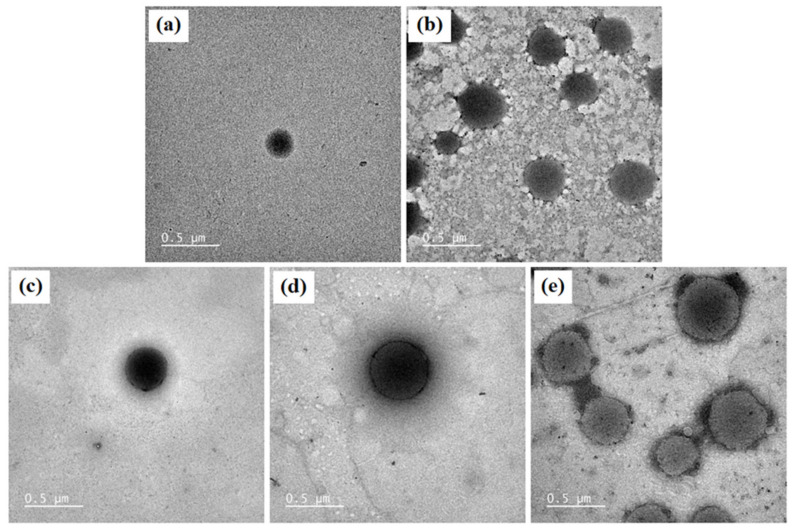
The morphology of (**a**) blank niosomes, (**b**) blank niosomes in F1, (**c**) MLT niosomes in F1, (**d**) S-MLT niosomes in F1, (**e**) and G-MLT niosomes in F1 under TEM. Scale bars = 500 nm.

**Figure 5 polymers-13-01142-f005:**
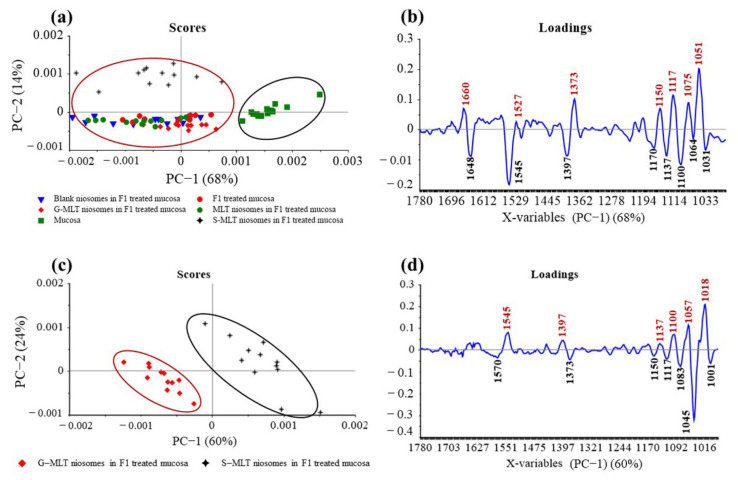
(**a**) Principal component (PC) plots and (**b**) loading plots of (●) F1, (▼) blank niosomes in F1, (●) MLT niosomes in F1, (✦) S-MLT niosomes in F1, and (◆) G-MLT niosomes in F1-treated mucosa compare to untreated mucosa (▪) at fingerprint region. (**c**) PC plots and (**d**) loading plots of (✦) S-MLT niosomes in F1-treated mucosa compared to (◆) G-MLT niosomes in F1-treated mucosa at fingerprint region.

**Figure 6 polymers-13-01142-f006:**
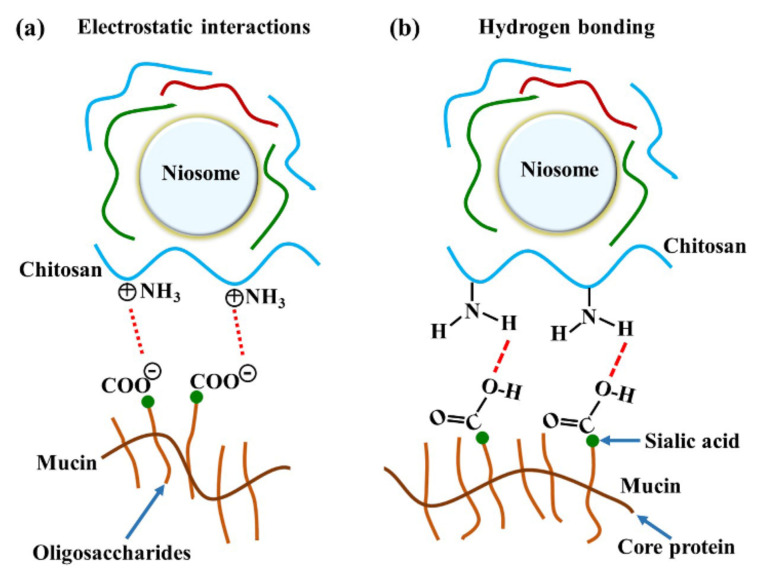
The illustration of chitosan and mucin interactions by (**a**) electrostatic interaction and (**b**) hydrogen bonding.

**Figure 7 polymers-13-01142-f007:**
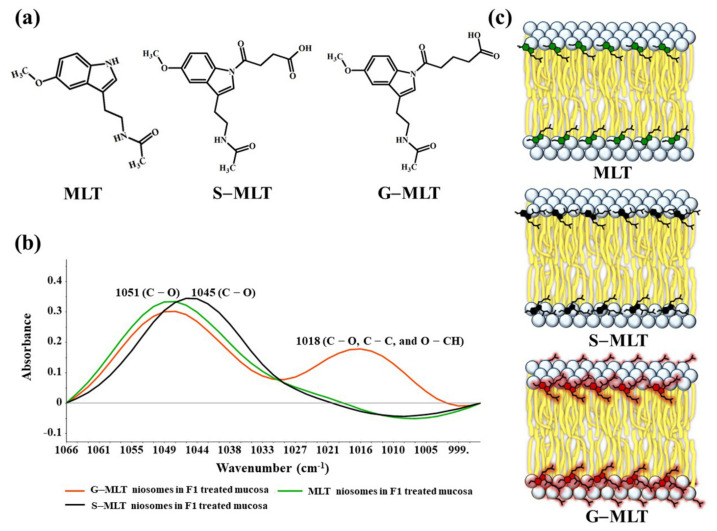
(**a**) The chemical structures of MLT, S-MLT, and G-MLT; (**b**) the ATR-FTIR spectra of MLT (green), S-MLT-(black), and G-MLT-(red) niosomes in F1-treated mucosa at wavenumber 1066–995 cm^−1^; (**c**) the schematic of MLT or its derivatives embedded in lipid bilayer of niosome.

**Figure 8 polymers-13-01142-f008:**
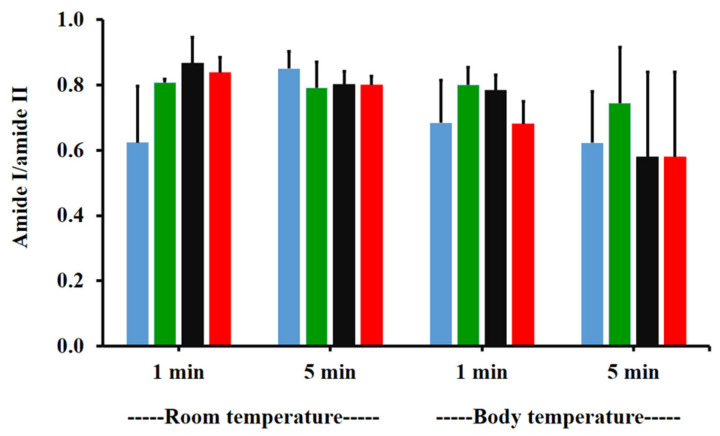
The amide I/amide II ratio of niosome gel (blank niosomes in F1 (blue), MLT niosomes in F1 (green), S-MLT niosomes in F1 (black), and G-MLT niosomes in F1 (red)) treated mucosa/untreated mucosa at BT and RT for 1 and 5 min. Experiments were performed in triplicate and repeated three times. Bars display mean ± S.D.

**Figure 9 polymers-13-01142-f009:**
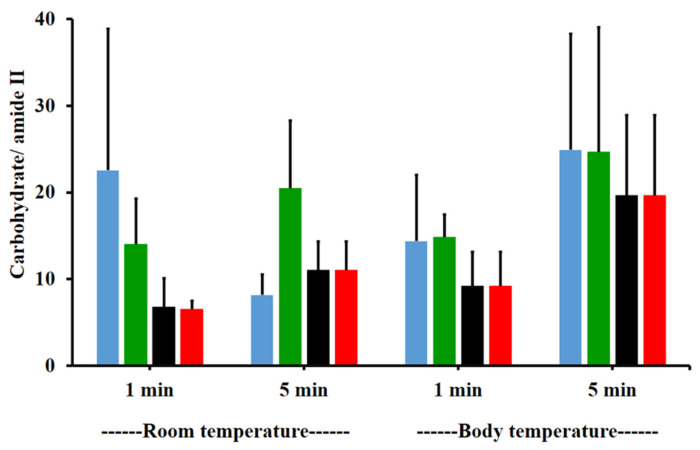
The carbohydrate/amide II ratio of niosome gel (blank niosomes in F1 (blue), MLT niosomes in F1 (green), S-MLT niosomes in F1 (black), and G-MLT niosomes in F1 (red)) treated mucosa to untreated mucosa at BT and RT for 1 and 5 min. Experiments were performed in triplicate and repeated three times. Bars display mean ± S.D.

**Table 1 polymers-13-01142-t001:** The characterizations of blank niosomes, gels, and niosome gels. Data are the mean ± SD.

	Blank Niosomes	Blank Niosomes in F1	MLT Niosomes in F1	S-MLT Niosomes in F1	G-MLT Niosomes in F1
**Physical parameters**					
Vesicle sizes (nm, *n* = 10)	331 ± 173	281 ± 55	414 ± 138 ^#^	535 ±130	314 ± 45
Polydispersity index (p.i.) (*n* = 6)	1.5 ± 1.2	2.1 ± 0.5	1.2 ± 0.6 ^#^	1.6 ± 0.7	1.3 ± 0.5 ^#^
Zeta potential (mV, *n* = 3)	−71.4 ± 1.0	−5.6 ± 0.4 *	−32.4 ± 2.9 *^#^	−38.2 ± 0.2 *^#^	−1.1 ± 0.4 *^#†§^
Encapsulation (%, *n* = 3)	ND	ND	92.63 ± 3.11	93.05 ± 2.87	88.99 ± 4.02
Permeation (*n* = 3) Fluxes (ug/cm^2^/h)	ND	ND	16.60 ± 5.51	0.08 ± 0.01^†^	0.29 ± 0.05 ^†§^
Lag time (h)	ND	ND	2.38 ± 0.05	5.15 ± 0.61^†^	2.44 ± 1.26 ^§^

* *p* < 0.05 vs. blank niosome, ^#^
*p* < 0.05 vs. Blank niosomes in F1, ^†^
*p* < 0.05 vs. MLT niosomes in F1, ^§^
*p* < 0.05 vs. S-MLT niosomes in F1; ND is not determined.

## Data Availability

Not applicable.

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
