# Peer review of "Characteristic Evaluation of Gel Formulation Containing Niosomes of Melatonin or Its Derivative and Mucoadhesive Properties Using ATR-FTIR Spectroscopy"

_polymers, 2021, doi:10.3390/polym13071142_

Round 1
Reviewer 1 Report
In the manuscript “Characteristic evaluation of gel formulation containing niosomes of melatonin or its derivative and mucoadhesive properties using ATR-FTIR spectroscopy” by Uthaiwat et colleagues, the Authors applied the FTIR approach to compare the mucoadhesive properties of previously studied gel formula with a gel containing HPMC, P407 and chitosan.
The issue is interesting for its applications, but, in general, the rationale behind the results is missing.
Major points:
Line 211: peptide bond should be better instead of amino-acid
Lines 241-243: where are the spectra of mucin treated with F1 and F2? If not shown, Authors should report them.
Lines 245-249: the Authors should explain better why the removal of proteins at the interface should modify the AI/AII ratio. This concept is not clear. In fact, you might expect a simultaneous variation – in the same direction - of both Amide I and Amide II.
Lines 249-252: Please, add some references to support this concept.
Figure 3: the reported ratios without the observation of the spectra are not significant. For example, the variation of the ratios should be due to a different hydration? Please, provide the measured spectra.
Lines 376-378: In which spectral range? If the Authors speak about NH2 and COOH, we are out of the Amide I band.
Lines 404-406: please, reformulate this sentence. The concept could be in some way understood, but that gels interact with Amide I and Amide II is a not correct concept….
In general, the Authors should report – also as supporting information – the measured spectra used to prepare the figures. This could help to support their results.
Minor point:
Line 75: with instead of bwith
Reviewer 2 Report
In the manuscript " Characteristic evaluation of gel formulation containing nio-2 somes of melatonin or its derivative and mucoadhesive proper-3 ties using ATR-FTIR spectroscopy " by ATR-FTIR spectroscopy, they report a comparative study of the in vitro mucoadhesive properties of their previous transmucosal formula containing HPMC, P407 and PVP with another formula containing HPMC, P407 and chitosan.
The overall manuscript is well written, the methodologies well described and the results supported. Even if I do not see any novelty in the presented work, however, is a comparative study, I recommend it for publication after minor revision.
1) Figure 2 The error bar represents the standard deviation? if yes, in the case of F2 treated mucid it is very wide, could you explain it? the number of the experiments should be defined in the caption.
2) In the introduction part more attention has to be paid to the reason to use ATR-FTIR for mucoadhesivity evaluation and the advantages compared to other approaches
3) Figure 8 and 9. The number of measurements should be mentioned in the caption
Round 2
Reviewer 1 Report
The authors have satisfactorily responded to all my questionsand made the necessary changes to the manuscript.
Only one minor point:
In Figure 1 (and other similar figures), the y axis should be
"Absorbance (arbitrary unit or a. u.)", instead of arbitrary unit.
Author Response
Thank you so much for all of your comments.
"Please see the attachment."
